

# A set of RT-PCR assays for detection of all known avian paramyxoviruses and application in surveillance of avian paramyxoviruses in China

Ji-Hui Jin[1], Jing-Jing Wang[1], Ying-Chao Ren[2], Shuo Liu[1], Jin-Ping Li[1], Guang-Yu Hou[1], Hua-Lei Liu[1], Qing-Ye Zhuang[1], Su-Chun Wang[1], Wen-Ming Jiang[1], Xiao-Hui Yu[1], Jian-Min Yu[1], Li-Ping Yuan[1], Cheng Peng[1], Guo-Zhong Zhang[3] and Ji-Ming Chen[1]

[1] Laboratory for Avian Disease Surveillance (OIE Reference Laboratory for Newcastle Disease), China Animal Health and Epidemiology Center, Qingdao, China
[2] Department for Animal Health Assessment, China Animal Health and Epidemiology Center, Qingdao, China
[3] Key Laboratory of Animal Epidemiology of the Ministry of Agriculture, College of Veterinary Medicine, China Agricultural University, Beijing, China

Corresponding author
Ji-Ming Chen, chenjiming@cahec.cn

## ABSTRACT

**Background:** Avian paramyxoviruses (APMVs), also termed avian avulaviruses, are of a vast diversity and great significance in poultry. Detection of all known APMVs is challenging, and distribution of APMVs have not been well investigated.
**Methods:** A set of reverse transcription polymerase chain reaction (RT-PCR) assays for detection of all known APMVs were established using degenerate primers targeting the viral polymerase L gene. The assays were preliminarily evaluated using in-vitro transcribed double-stranded RNA controls and 24 known viruses, and then they were employed to detect 4,346 avian samples collected from 11 provinces.
**Results:** The assays could detect 20–200 copies of the double-stranded RNA controls, and detected correctly the 24 known viruses. Of the 4,346 avian samples detected using the assays, 72 samples were found positive. Of the 72 positives, 70 were confirmed through sequencing, indicating the assays were specific for APMVs. The 4,346 samples were also detected using a reported RT-PCR assay, and the results showed this RT-PCR assay was less sensitive than the assays reported here. Of the 70 confirmed positives, 40 were class I Newcastle disease virus (NDV or APMV-1) and 27 were class II NDV from poultry including chickens, ducks, geese, and pigeons, and three were APMV-2 from parrots. The surveillance identified APMV-2 in parrots for the first time, and revealed that prevalence of NDVs in live poultry markets was higher than that in poultry farms. The surveillance also suggested that class I NDVs in chickens could be as prevalent as in ducks, and class II NDVs in ducks could be more prevalent than in chickens, and class II NDVs could be more prevalent than class I NDVs in ducks. Altogether, we developed a set of specific and sensitive RT-PCR assays for detection of all known APMVs, and conducted a large-scale surveillance using the assays which shed novel insights into APMV epidemiology.

# INTRODUCTION

Avian paramyxoviruses (APMVs), newly designated as avian avulaviruses (AAvVs), belong to the *Avulavirinae* subfamily of *Paramyxoviridae*. So far, as given in Table 1, three genera (*Orthoavulavirus*, *Metaavulavirus*, *Paravulavirus*) and 20 species of APMVs (APMV-1 to APMV-20) have been identified (*Aziz-ul-Rahman, Munir & Shabbir, 2018*; *International Committee on Taxonomy of Viruses, 2019*). Of the 20 species, infections of APMV-1, APMV-2, APMV-3, APMV-6, and APMV-7 can cause avian morbidity (*Awang & Russell, 1990*; *Bankowski, Almquist & Dombruski, 1995*; *Lipkind et al., 1995*; *Woolcock et al., 1996*; *Saif et al., 1997*; *Shihmanter et al., 2000*). In particular, APMV-1, which is usually termed Newcastle disease virus (NDV), is the pathogen of Newcastle disease, an acute, highly contagious infectious disease that infects a variety of avian species (*Dimitrov et al., 2019*; *Zhan et al., 2020*). NDVs are divided into class I and class II NDVs, and both classes have evolved into multiple genotypes (*Ramey et al., 2013*; *Dimitrov et al., 2016, 2019*; *Hicks et al., 2019*).

Various reverse transcription polymerase chain reaction (RT-PCR) assays have been established for detection of APMVs (*Liu et al., 2011*; *Fornells et al., 2013*; *Sutton et al., 2019*). Most of these assays were designed with primers specific to one species, and thus only one species of APMVs could be detected with one of these assays. Moreover, these assays might be unable to detect the seven species ranging from APMV-14 to APMV-20, which were identified in recent years and differ greatly from other species in gene sequences (*Karamendin et al., 2017*; *Lee et al., 2017*; *Neira et al., 2017*; *Thampaisarn et al., 2017*; *Thomazelli et al., 2017*). Therefore, it is highly desired to develop novel assays for detection of all known species of APMVs.

The genomic sequences of APMVs are highly diversified, and comparative analysis revealed that the whole genome sequences could differ at over 65% sites among APMVs (*Aziz-ul-Rahman, Munir & Shabbir, 2018*). The extensive variability stems mainly from nucleotide substitutions, insertions, and deletions, causing challenges in developing a universal assay for detection of AMPVs based on their genomic sequences.

The L gene in the APMV's genome encodes the RNA polymerase protein. This gene is the most conserved in the viral genome, and its sequence differences are the criteria for the delineation of APMVs species (*Aziz-ul-Rahman, Munir & Shabbir, 2018*; *Rima et al., 2018*). We designed in this study four pairs of degenerate primers according to the conserved region of the L gene of representative strains of 20 species of APMVs, with the goal to establish a set of RT-PCR assays capable of detecting all APMVs. We further applied these assays in surveillance of APMVs in China.

# MATERIALS AND METHODS

## Ethics statement

All procedures involving animals were performed in accordance with regulatory standards and guidelines approved by the Animal Care and Use Committee of China Animal Health

**Table 1 Four pairs of degenerate primers for detection of avian paramyxoviruses.**

| Pair | Primer sequence[a] | Target genus | Target species | Amplicon size | Control's size |
|------|-------------------|--------------|----------------|---------------|----------------|
| 1 | 1/3F: AAGTACTGTCTTAAYTGGAGRTA<br>1R: ACTTGATTGTCACCYTGYACCAT | *Orthoavulavirus* | APMV-1, -9, -12, -13, -16, -17, -18, -19 | 332 bp | 420 bp |
| 2 | 2F: GTCTCATACTCACTCAAAGAGAA<br>2R: GGGTCTGCAACRTACATRGTG | *Paravulavirus* | APMV-3, -4 | 416 bp | 370 bp |
| 3 | 1/3F: AAGTACTGTCTTAAYTGGAGRTA<br>3R: CATAGTCCACATYTTYTGRCATA | *Metaavulavirus* | APMV-5, -6, -7, -14 | 252 bp | 310 bp |
| 4 | 4F: AATGGAGTGTCRATGGARCA<br>4R: CGCTAATTGADATCATDGTCCA | *Metaavulavirus* | APMV-2, -8, -10, -11, -15, -20 | 490 bp | 400 bp |

**Note:**
[a] R = A/G, Y = T/C, D = A/T/G.

and Epidemiology Center (No. 2018LSAD-05). Swab and feces samples were collected with permission granted by multiple relevant parties, including China Animal Health and Epidemiology Center, the veterinary administration of local governments, and the relevant farm owners (No. 2018LSAD-FD05).

## Design and synthesis of primers

Nucleotide sequences of the L gene of 49 strains covering all the known genera and species of APMVs were downloaded from GenBank and aligned using the software package MEGA 7.0 for search of conserved regions. As per the conserved regions, four pairs of degenerate primers were designed for detection of the known genera and species of APMVs (Table 1).

## Preparation of RNA controls

Double strand RNA (dsRNA) is stable than single strand RNA, and thus dsRNA was employed as the RNA control for the assays, as described previously (*Chen et al., 2006*). Briefly, to differentiate potential false positives caused by contamination of positive controls, sequences of the positive controls were different in length from the target regions of the RT-PCR assays (Table 1), through inserting or deleting some nucleotides flanking the binding sites of the upstream and downstream primers in the positive controls. The positive control sequences were shown in the Table S1.

The DNA sequences of the positive controls were synthesized and cloned into pUC57 vector. The constructed vectors were amplified using the vector-specific primers, pUC57F (5′-TAATACGACTCACTATAGGGGACTGCAGAGGCCTGCATGC-3′) and pUC57R (5′-TAATACGACTCACTATAGGGACCATGATTACGCCAAGCTT-3′). Primers pUC57F and pUC57R contained the T7 promoter at the 5′ end. The PCR products were purified using a QIAquick column extraction kit (Qiagen, Hilden, Germany). The purified PCR products were transcribed using the T7 in vitro transcription kit (Takara, Dalian, China). The transcribed product was digested with Exonuclease III (Takara, Dalian, China) to remove the DNA template, and then purified with the RNAeasy purification kit (Qiagen, Hilden, Germany). The dsRNA concentration was measured using the NanoDrop ND-100 spectrophotometer (Nano Drop, Delaware, USA).

## Establishment of the assays

Using the designed primers, dsRNA controls, and reverse transcription polymerase chain reaction (RT-PCR), a set of four RT-PCR assays for detection of all APMVs were optimized regarding primer concentration, annealing temperature, and other cycling parameters. The assays were performed in a 25 μL reaction volume using PrimeScript™ One Step RT-PCR Kit Ver.2 (Takara, Dalian, China) as follows: 12.5 μL buffer, 2.0 μL primer mixture (each 2 mM, final), 1.0 μL enzyme mixture, 3.0 μL template, and 6.5 μL RNase Free $H_2O$. The reaction was conducted with an initial reverse transcription step at 50 °C for 30 min, followed by PCR activation at 95 °C for 2 min, 35 cycles of amplification (30 s at 95 °C, 30 s at 55 °C, 30 s at 72 °C), and a final extension step at 72 °C for 5 min. The RT-PCR products were then subjected to capillary electrophoresis using the QIAxcel analyzer (Qiagen, Hilden, Germany).

## Preliminary evaluation of the RT-PCR assays

Two avian influenza viruses (AIVs), one infectious bronchitis virus, one infectious bursal disease virus (IBDV), and twenty NDV strains were propagated in 10-day-old embryonated specific-pathogen-free (SPF) chicken eggs in our laboratory. Their RNA was extracted from 100 μL of allantoic fluid with the RNeasy Mini Kit (Qiagen, Hilden, Germany), for detection of specificity of the RT-PCR assays.

## Application of the developed assays in APMV surveillance

We conducted this APMV surveillance as per the methods of the surveillance we published previously (*Jiang et al., 2012*). Briefly, 4,346 avian samples were collected from 11 provinces of China, including 4,122 poultry swab samples from chickens, ducks, geese (*Anser cygnoides domestica*), pigeons, and newly domesticated bar-headed geese (*Answer indicus*). The 4,346 samples also included 224 feces samples collected from parrots (*Melopsittacus undulatus*), thrushes (*Garrulax canorus*), mynahs (*Acridotheres cristatellus*), skylarks (*Alauda arvensis*), and larks (*Eremophila alpestris*). These 4,346 samples were collected from 12 live bird wholesale markets, 42 live bird retail markets, six poultry farms, two backyard flocks, four slaughtering houses, and three pet bird markets in late 2018 and early 2019, mainly for surveillance of AIVs and NDVs circulating in poultry in China. The host species, provinces, and sites where these samples were collected were given in the Data S1.

Nylon flocked swabs were employed to take smears at both cloacal and oropharyngeal tracts of a bird, and the swab samples were stored in 1.5 mL phosphate-buffered saline (PBS) containing 10% glycerol. Feces samples were collected by taking approximately 0.5 mL fresh wet feces, and stored in 3.5 mL PBS containing 10% glycerol. The samples were stored at 4 °C and detected in 72 h after collection.

The samples were centrifuged at 1,000 g for 5 min, and the supernatants were inoculated in 10-day-old embryonated SPF chicken eggs via the allantoic sac route. The eggs were further incubated for 4 d and checked twice each day. Dead ones were removed and stored at 4 °C. Then, the allantoic fluids of the live embryos were collected and tested using the hemagglutination assay in order to test for the presence of viruses. The RNA of all

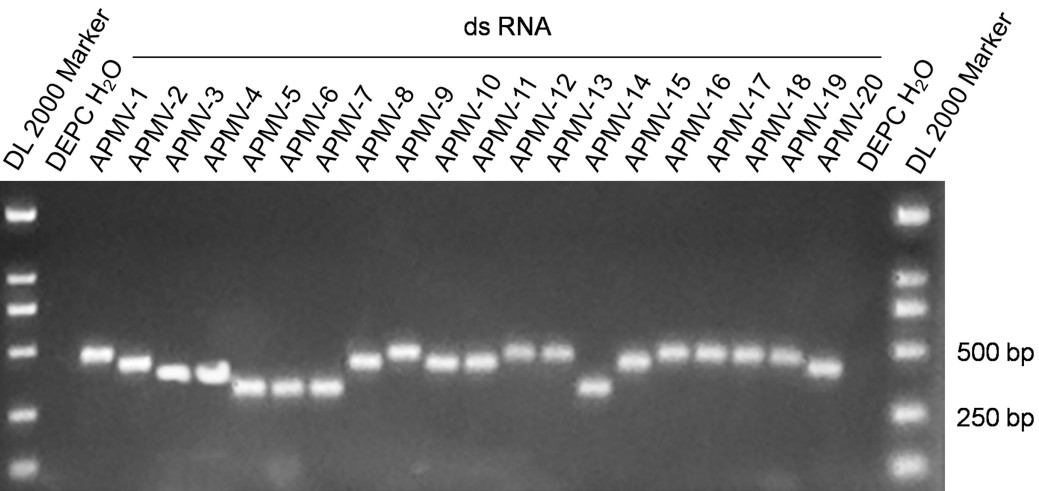

**Figure 1 Electrophoresis of the amplification products of the 20 dsRNA positive controls.**

hemagglutination-positive allantoic fluids of live eggs, allantoic fluids of all the dead eggs, and all feces samples was extracted by using the RNeasy Mini kit (Qiagen). The extracted RNA was used for detection of AIVs, NDVs, IBVs, and APMVs using relevant RT-PCR assays, and all positive amplicons were sequenced for confirmation and molecular epidemiological analysis. The GenBank accession numbers for the sequences originally reported in this study are MT738103–MT738172.

## Phylogenetic analysis of viral sequences

Viral sequences were analyzed using the Software packages of Mega 7.0. Briefly, sequences were aligned using the software Clustal X. Then, the phylogenetic relationships were calculated using the maximum-likelihood and the model of K2P+G+I (Kimura 2-parameter plus gamma distribution with a fraction of evolutionary invariable sites), because this model is considered to describe the nucleotide substitution pattern the best since it has the lowest Bayesian Information Criterion (*Kumar, Stecher & Tamura, 2016*). Bootstrap values were calculated out of 1,000 replicates. Genetic distances were calculated using the same model and parameters set for phylogenetic analysis.

## RESULTS

### Preparation of the dsRNA positive controls

The 20 dsRNA positive control standards were produced at the microgram level through in vitro transcription. The absorbance ratio of OD260 nm to OD280 nm of the RNA transcripts was in the range of 2.0 ± 0.1, which indicated the high quality of the RNA positive controls. Figure 1 shows the results of agarose gel electrophoresis of the 20 dsRNA controls with the expected sizes.

### Preliminary evaluation of the assays

Using the serial dilutions of the dsRNA controls, the developed assays, after optimized in primer concentration, annealing temperature, and other cycling parameters, could detect

**Table 2 Detection of 72 positives with the four pairs of primers.**

| Sequencing confirmed | First pair | Second pair | Third pair | Fourth pairs |
|---|---|---|---|---|
| 67 NDV | 67 | 1 | 4 | 2 |
| 3 APMV2 | 0 | 0 | 0 | 3 |
| 2 unknown | 2 | 0 | 0 | 0 |

20–200 of the dsRNA copies. Detection of the extracted RNA of two AIVs, one IBV, one IBDV, and twenty NDVs with the assays only found that the RNA from the 20 NDVs were positive. These data indicated that the assays could be sensitive and specific.

## Application of the assays for surveillance of APMVs

The 4,346 samples were tested using the set of four RT-PCR assays developed in this study, and 72 samples were positive detected with at least one pair of the primers (Table 2). Of these 72 positives, 70 positives were confirmed through sequencing and BLAST analysis of the sequences, and we did not obtain reliable sequences from the remaining two positive amplicons. This suggested that the set of RT-PCR assays were highly specific (the specificity which is the possibility that an authentically negative sample to be detected as a positive sample ≥ 1−2/(4,346−70) = 99.95%). A few NDV-positive samples were also detected with the primers designed for other species of APMVs (Tables 1 and 2). The 4,346 samples were also detected using a reported RT-PCR assay targeting the F gene of all NDVs (*Liu et al., 2011*), and 55 positives were found, and 52 of these 55 samples were also found positive in the set of RT-PCR assays developed in this study (see the Data S2). These data suggested that our RT-PCR assays were significantly higher in sensitivity than the reported RT-PCR assay targeting all NDVs ($P < 0.01$, by the Chi-square test).

Of the 70 APMV positives, 67 APMV-1 or NDV positives which were all from poultry, and three APMV-2 positives which were all from the parrots in pet bird markets. We checked all sequences of APMV-2 in GenBank, and found that this study identified APMV-2 in parrots for the first time. The 224 feces samples were detected twice, with and without inoculation in embryonated eggs. All the three APMV-2 positives were detected without inoculation in embryonated eggs, and only one of the three APMV-2 positives were detected with inoculation in embryonated eggs.

No samples from poultry farms were found NDV positive by the assays, and one or more positive NDV samples from backyard flocks, live wholesale poultry markets, and live retail poultry markets were found (Table 3). The NDV-positive rate in the samples from live retail poultry markets was significantly higher than that in the samples from live wholesale poultry markets ($P < 0.01$, by the Chi-square test), which was further higher than that in the samples from poultry farms ($P < 0.01$, by the Chi-square test).

Phylogenetic relationships among the sequences of the amplicons of the NDV positives were analyzed along with some standard sequences used for universal classification of NDVs (*Dimitrov et al., 2019*). The results suggested that the amplicon sequences could be employed to clearly differentiate class I NDVs and class II NDVs (Fig. 2). Among the

**Table 3 NDV positives identified in the samples from different poultry sites.**

|  | Poultry farms | Live wholesale markets | Live retail markets | Slaughtering houses | Backyard flocks |
|---|---|---|---|---|---|
| No. samples | 360 | 1,195 | 2,175 | 337 | 55 |
| No. positives | 0 | 11 | 51 | 4 | 1 |
| Prevalence (%) | 0.00 | 0.92 | 2.34 | 1.19 | 1.82 |

67 NDV positives, 40 were class I NDVs and 27 were class II NDVs, and their prevalence in different species of birds was given in Table 4. As mentioned above, 52 of the 67 NDV positives were also found NDV positive with a previously reported RT-PCR assay targeting the viral F gene (*Liu et al., 2011*). Classification of these 52 NDVs into the two classes using the amplicon sequences of the assays reported here was the same as that using the amplicon sequences of the assay reported previously.

It was assumed in the past that class I NDVs mainly circulate in wild birds and waterfowls, and class II NDVs in ducks are less prevalent than class II NDVs in chickens and class I NDVs in ducks (*Ramey et al., 2013*; *Dimitrov et al., 2016*, *2019*; *Hicks et al., 2019*). Interestingly, here we showed that class I NDVs in chickens were as prevalent as in ducks ($P > 0.05$, by the Chi-square test), and that class II NDVs in ducks was significantly more prevalent than class II NDVs in chickens and class I NDVs in ducks ($P < 0.01$, by the Chi-square test). These data suggested that ducks could be the asymptomatic reservoir of class II NDVs, some of which could cause severe disease in chickens (*Wu et al., 2015*).

## DISCUSSION

In this study, we designed a set of four RT-PCR assays for detection of all known APMVs, and prepared dsRNA controls for the assays. These assays were preliminarily evaluated using the dsRNA and some known avian viruses. Then these assays were employed in a large-scale surveillance of avian diseases including APMVs. The surveillance results demonstrated that the assays were highly specific and more sensitive than a reported RT-PCR targeting the viral F gene. The surveillance results also provided novel important epidemiological information pertaining to APMVs in China.

Because the genome sequences of APMVs are highly variable, we tried but failed to design a single pair of primers for detection of all known APMVs. A conserved RT-PCR assay for detection all paramyxoviruses was reported (*Van Boheemen et al., 2012*), but we analyzed on computer and found that the conserved RT-PCR assay is suitable for many paramyxoviruses, but not for some species of APMVs. A set of semi-nested or nested RT-PCR assays targeting the viral L gene sequences had been established for detection of all paramyxoviruses (*Tong et al., 2008*). These assays could sensitively detect APMVs. However, the assays require two steps of amplification and opening of the amplification system. They are thus time-consuming and risky in nucleic acid contamination. Our RT-PCR assays only require one step of amplification without opening of the amplification system, and are thus more suitable for APMV detection and surveillance.

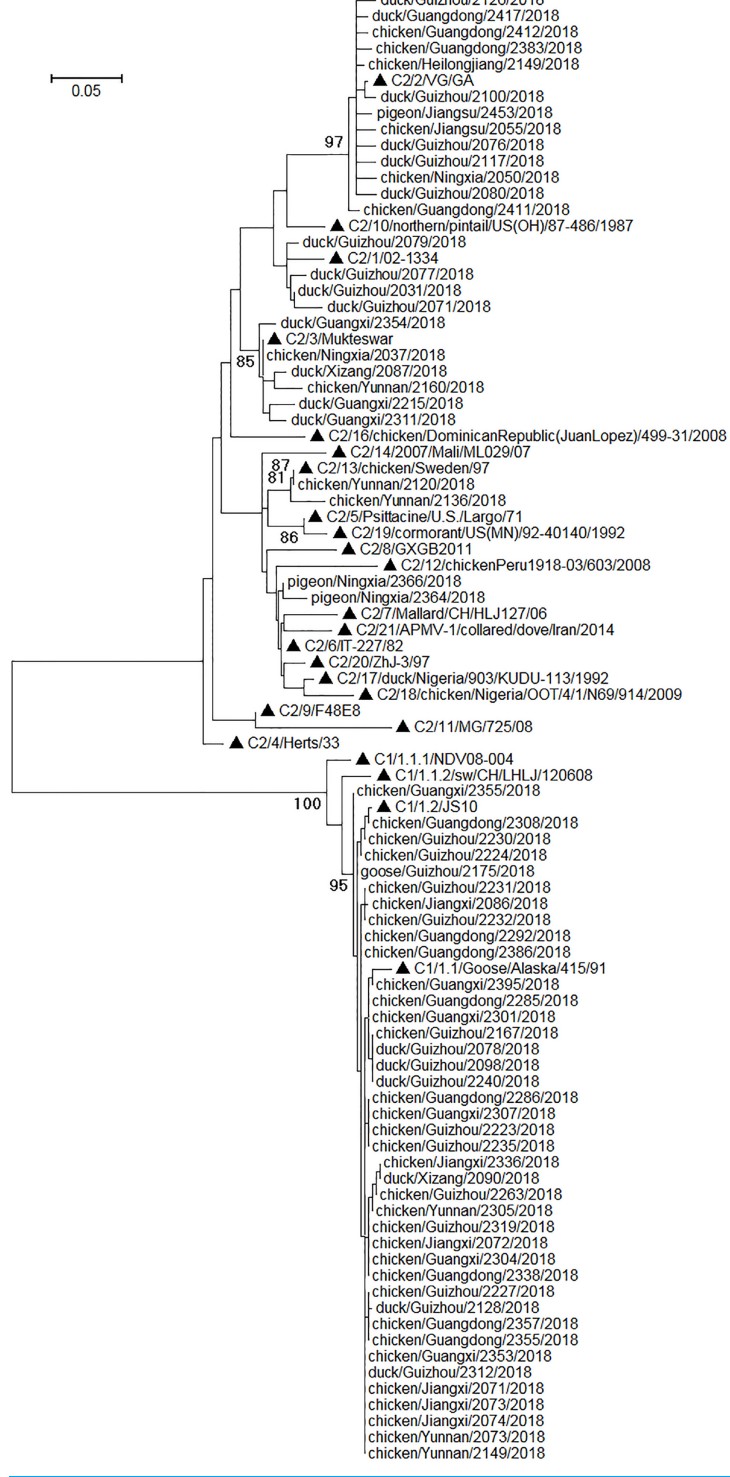

**Figure 2 Phylogenetic relationships among 67 NDVs identified in this study and 24 reference strains for the universal NDV classification.** The reference strains were marked with triangles and initiated with their nomenclature (e.g., C1/1.2/ means class I genotype 1.2, and C2/10/ means class II genotype X). The scale bar indicated genetic distances. The viruses identified in this study were designated with the corresponding bird species, sample collection locations, sample numbers, and sample collection years.

**Table 4 NDV positives identified in the samples from different species of birds.**

|  | Chickens (n = 3,145) | Ducks (n = 616) | Geese (n = 58) | Pigeons (n = 293) | Others (n = 234) |
|---|---|---|---|---|---|
| Class I | 33 (1.05%) | 6 (0.97%) | 1 (1.72%) | 0 (0.00%) | 0 (0.00%) |
| Class II | 10 (0.32%) | 14 (2.27%) | 0 (0.00%) | 3 (1.02%) | 0 (0.00%) |

The dsRNA positive controls used in our RT-PCR assays have advantages over normal RNA controls used in RT-PCR, because they are stable and can differentiate false positives caused by nucleic acid contamination.

The large-scale surveillance conducted in this study updated our knowledge pertaining to the epidemiology of APMVs. First, it demonstrated that class I NDVs in chickens could be as prevalent as in ducks, and class II NDVs could be more prevalent in ducks than in chickens, and class II NDVs could be more prevalent than class I NDVs in ducks, all of which are different from our previous views (*Ramey et al., 2013*; *Dimitrov et al., 2016*, *2019*; *Hicks et al., 2019*). Second, the surveillance suggested that the prevalence of NDV increased during their transportation from poultry farms to live wholesale poultry markets and then to live retail poultry markets. This indicated that live poultry markets are the hot sites not only for transmission and replication of AIVs (*Liu et al., 2020*), but also for transmission and replication of NDVs. Third, the surveillance identified for the first time that APMV-2 circulated in parrots which are popular pet birds worldwide. We will analyze the genomic features of the APMVs and investigate their pathogenesis in parrots in the coming future.

The surveillance was of some limitations. First, the tested samples were collected mainly for surveillance of AIVs and NDVs, and they were hence incubated with embryonated eggs. Some APMVs could be undetected through this way because they grow inefficiently in embryonated eggs or have no hemagglutination activity, as suggested by the detection of AMPV-2 in this study and a hemagglutination-negative APMV-4 strain we identified in 2013 (*Wang et al., 2013*). Second, only a few poultry farms, backyard flocks, slaughtering houses, and pet bird markets were detected, and thus the results could not reflect the prevalence of APMVs in these types of poultry sites. Meanwhile, samples from geese, and pigeons were also inadequate to reflect the prevalence of APMVs in these species of birds. We will circumvent these limitations for performing surveillance of APMVs in the future.

## CONCLUSION

In conclusion, we developed a set of specific and sensitive RT-PCR assays for detection of all known APMVs, and conducted a large-scale surveillance using the assays which shed novel insights into to APMV epidemiology in China.

### Funding

This study was supported by the Postdoctoral Innovative Talents Support Program of Shandong Province and the Qingdao Postdoctoral Applied Research Project. The funders

had no role in study design, data collection and analysis, decision to publish, or preparation of the manuscript.

## Grant Disclosures

The following grant information was disclosed by the authors:
Postdoctoral Innovative Talents Support Program of Shandong Province.
Qingdao Postdoctoral Applied Research Project.

## Competing Interests

The authors declare that they have no competing interests.

## Author Contributions

- Ji-Hui Jin performed the experiments, analyzed the data, prepared figures and/or tables, authored or reviewed drafts of the paper, applied successfully the relevant funds, and approved the final draft.
- Jing-Jing Wang performed the experiments, prepared figures and/or tables, and approved the final draft.
- Ying-Chao Ren performed the experiments, prepared figures and/or tables, and approved the final draft.
- Shuo Liu performed the experiments, prepared figures and/or tables, and approved the final draft.
- Jin-Ping Li performed the experiments, prepared figures and/or tables, and approved the final draft.
- Guang-Yu Hou performed the experiments, prepared figures and/or tables, and approved the final draft.
- Hua-Lei Liu performed the experiments, prepared figures and/or tables, and approved the final draft.
- Qing-Ye Zhuang performed the experiments, prepared figures and/or tables, and approved the final draft.
- Su-Chun Wang performed the experiments, prepared figures and/or tables, and approved the final draft.
- Wen-Ming Jiang performed the experiments, prepared figures and/or tables, and approved the final draft.
- Xiao-Hui Yu performed the experiments, prepared figures and/or tables, and approved the final draft.
- Jian-Min Yu performed the experiments, prepared figures and/or tables, and approved the final draft.
- Li-Ping Yuan performed the experiments, prepared figures and/or tables, and approved the final draft.
- Cheng Peng performed the experiments, prepared figures and/or tables, and approved the final draft.
- Guo-Zhong Zhang performed the experiments, analyzed the data, prepared figures and/or tables, and approved the final draft.

- Ji-Ming Chen conceived and designed the experiments, performed the experiments, analyzed the data, prepared figures and/or tables, authored or reviewed drafts of the paper, applied successfully the relevant funds, and approved the final draft.

## Animal Ethics

The following information was supplied relating to ethical approvals (i.e., approving body and any reference numbers):

The Animal Care and Use Committee of China Animal Health and Epidemiology Center provided full approval for this research (No. 2018LSAD-05).

## Field Study Permissions

The following information was supplied relating to field study approvals (i.e., approving body and any reference numbers):

China Animal Health and Epidemiology Center, the veterinary administration of local governments, and the relevant farm owners approved the field experiments (No. 2018LSAD-FD05).

## DNA Deposition

The following information was supplied regarding the deposition of DNA sequences:

The viral sequences originally reported in this study are available in the Supplemental Files and at GenBank: MT738103 to MT738172.

## Data Availability

Raw data are available in the Supplemental Files.

## Supplemental Information

Supplemental information for this article can be found online at http://dx.doi.org/10.7717/peerj.10748#supplemental-information.

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
