# Peer review of "A set of RT-PCR assays for detection of all known avian paramyxoviruses and application in surveillance of avian paramyxoviruses in China"

_PeerJ, doi:10.7717/peerj.10748_

## Round 0.1 · original submission · Major Revisions

Please check the reviewers' comments and revise them accordingly.

·

Basic reporting

The article is written in clear, unambiguous language. Sufficient background and context is provided. The structure of the article is professional.

Experimental design

The research question is well defined. Ethical standards are met (including the use of animals). The methods are described in sufficient detail.

Validity of the findings

It would be beneficial to provide all of the surveillance data as opposed to only the positive. The negatives are likely informative as well.

I do have a concern about the specificity of the developed assays described in the manuscript. How can the authors establish the specificity of these assays? It does not seem like the authors have described this.

·

Basic reporting

The manuscript had generally good writing and presentation of background.

1. The Introduction should explain class I and class II NDVs. No mention of these two classes are made until Line 211 of the Results.

2. Line 195: To clarify, when you say “poultry”, which animals do you mean? Only chickens? Or chickens, ducks, and geese, but not pigeons? Perhaps cite Table 3 here. Also clarify what species are in poultry farms and markets.

3. Figure 2: What does the scale 0.05 mean?

4. Figure 2 caption: The sample labels appear to indicate species/location/___/year – I don’t know what the third item is. For example, “2126” in “Duck/Guizhou/2126/2018”.

5. Line 264: I’m not sure what you mean by “the surveillance of APMVs hitchhiked the surveillance for AIVs and NDVs”. Do you mean the APMV assay was done as a supplement on samples which were collected primarily for AIV and NDV detection?

6. Line 248: I suggest “dsRNA” instead of “RNA”.

8. Line 155: consider adding, “…using the hemoagglutination assay in order to test for the presence of virus” (or another appropriate explanation).

9. Perhaps mention in abstract that L gene is RNA polymerase

Some other small language corrections:
Line 37: I suggest saying "this RT-PCR assay" instead of "this assay".
Line 134: space in "assayin"
Line 203: "positives" -> "positive"
Line 241: I suggest "had been established" instead of "were established"
Line 251: “recognition” -> “knowledge”
6. Line 253: space in "NDVscould"

Experimental design

I thought the methods and question of the study were explained well.

1. Line 189: When you say “failed in sequencing”, do you mean that these positives were found to not actually be NDV? Or was there a technical failure during sequencing?

2. Line 193: “…our RT-PCR assays were significantly higher than…” What does this mean? Maybe the accuracy or sensitivity was significantly higher?

Validity of the findings

The conclusions seemed to match the data. The data files looked fine to me.

---

## Round 0.2 · accepted · Accept

Please check the reviewer's comment and revise it accordingly.

·

Basic reporting

I'm satisfied with the changes the authors have made following my earlier suggestions.

Experimental design

I'm satisfied with the changes the authors have made following my earlier suggestions.

Small correction: line 102 , "more stable"

Validity of the findings

I'm satisfied with the changes the authors have made following my earlier suggestions.